# Language and Mental Health:
# Measures of Emotion Dynamics from Text as Linguistic Biosocial Markers

**Daniela Teodorescu**[1,2][*]**, Tiffany Cheng**[3]**, Alona Fyshe**[1,4]**, Saif M. Mohammad**[5]

[1]Dept. Computing Science, Alberta Machine Intelligence Institute (Amii), University of Alberta
[2]MaiNLP, Center for Information and Language Processing, LMU Munich, Germany
[3]Carleton University
[4]Dept. Psychology, University of Alberta
[5]National Research Council Canada

`{dteodore,alona}@ualberta.ca`, `tiffany.cheng@carleton.ca`, `saif.mohammad@nrc-cnrc.gc.ca`

## Abstract

Research in psychopathology has shown that, at an aggregate level, the patterns of emotional change over time—emotion dynamics—are indicators of one's mental health. One's patterns of emotion change have traditionally been determined through self-reports of emotions; however, there are known issues with accuracy, bias, and ease of data collection. Recent approaches to determining emotion dynamics from one's everyday utterances addresses many of these concerns, but it is not yet known whether these measures of *utterance emotion dynamics (UED)* correlate with mental health diagnoses. Here, for the first time, we study the relationship between tweet emotion dynamics and mental health disorders. We find that each of the UED metrics studied varied by the user's self-disclosed diagnosis. For example: average valence was significantly higher (i.e., more positive text) in the control group compared to users with ADHD, MDD, and PTSD. Valence variability was significantly lower in the control group compared to ADHD, depression, bipolar disorder, MDD, PTSD, and OCD but not PPD. Rise and recovery rates of valence also exhibited significant differences from the control. This work provides important early evidence for how linguistic cues pertaining to emotion dynamics can play a crucial role as biosocial markers for mental illnesses and aid in the understanding, diagnosis, and management of mental health disorders.

## 1 Introduction

Language is inherently *social*—from the way in which we say things, the expressions we use and the things we choose to share, being impacted by our social environment and lived experiences. As our social environments have evolved over time, language has evolved to better support our communication needs and collaborative societies. Therefore, language is also *variable*, as the way in which

we use it has adapted to cultures and communities around the world, and it is influenced by an individual's experiences.

Given the prominent role of language in human evolution from hunters–gathers to collaborative societies, and the large extent to which we rely on language today, it is not surprising that our mental health impacts our language usage. Quantitative features in language (e.g., aspects which can be measured) have already been shown to indicate and help clinicians monitor the progression of mental health conditions (MHCs), acting as *biomarkers*. A *linguistic biomarker* is a language-based measure that is associated with a disease outcome or biology in general (Ballman, 2015; Gagliardi and Tamburini, 2022; Lena, 2021). Some well-known linguistic biomarkers include: the proportion of pronouns (indicator of depression, Koops et al. (2023)), syntax reduction (Anorexia Nervosa, Cuteri et al. (2022)), certain lexical and syntactic features (mild cognitive impairment and dementia, Calzà et al. (2021); Gagliardi and Tamburini (2021)), and semantic connectedness (schizophrenia, Corcoran et al. (2020). Also, the emotions expressed in text have been shown to correlate with mental health diagnosis. For example, more negative sentiment in text by individuals with depression (De Choudhury et al., 2013; Seabrook et al., 2018; De Choudhury et al., 2021). Other work has shown that suicide watch, anxiety, and self-harm subreddits had noticeably lower negative sentiment compared to other mental health subreddits such as Autism and Asperger's (Gkotsis et al., 2016).

While language can be a biomarker for mental health, the substantial social nature of language has implications. Notably, the tremendous variability in language use—especially across social groups—means that we should be skeptical about universal biomarkers; and instead realize that linguistic biomarkers alone are not capable of predicting MHCs. A vast amount of contextual and

---

[*] Work done while at the University of Alberta.

clinical information (often only available to an individual's physician) helps determine well-being, and sometimes linguistic markers can aid the process. Further, linguistic biomarkers are more likely to be a stronger indicator among groups with commonalities; for example, when applied to people from the same region, culture, or medium of expression (e.g., social media platform). For example, social factors such as parental socioeconomic status, neighbourhood, and institutionalization (e.g., group foster care by government) influence speech markers such as lexical diversity; and social class influences syntactic complexity (Lena, 2021). Therefore, it is more appropriate to consider language as a *biosocial* marker for health as it is influenced by both social and biological factors (Lena, 2021).

As language is increasingly being studied as a biosocial marker for mental health – accelerated by the ease and availability of NLP tools and language data online – there are important ethical implications. We must consider the sociolinguistic factors of such markers to ensure less biased and more accessible tools in clinics (Lena, 2021). If social factors are not considered, then this limits the utility of systems and their ability to predict well-being as they may be capturing confounding variables.

In that context, our goal is to understand the extent to which patterns of emotion change act as *biosocial* markers for mental health? *Emotion dynamics* studies the patterns with which emotions change across time (Kuppens and Verduyn, 2015, 2017). Emotion dynamics have been shown to correlate with overall well-being, mental health, and psychopathology (the scientific study of mental illness or disorders) (Kuppens and Verduyn, 2017; Houben et al., 2015; Silk et al., 2011; Sperry et al., 2020). Further, studying emotion dynamics allows us to better understand emotions, and has been shown to have ties with academic success (Graziano et al., 2007; Phillips et al., 2002), and social interactions (e.g., shyness) in children (Sosa-Hernandez et al., 2022).

Emotion dynamics have been measured in psychology through self-report surveys over a period of time (e.g., 3 times a day for 5 days). Using these self-reports of emotion over time, various metrics can quantify how emotions change over time (e.g., the *average intensity*, *variability*, etc.). However, there are several drawbacks of using self-report surveys (which we discuss at the end of Section 2.1). Another window through which emotion dynamics

can be inferred is through one's everyday utterances. Hipson and Mohammad (2021) proposed the *Utterance Emotion Dynamics (UED)* framework which determines emotion dynamics from the emotion expressed in text. There are several metrics in the UED framework inspired by those in psychology (e.g., average emotion, emotion variability, rise rate, and recovery rate). Ties between emotion dynamics metrics and mental health have been shown in psychology, however it is not known whether this relationship similarly exists between emotion dynamics in one's *utterances/language* and mental health.

In this paper, we examine whether UED act as *biosocial* markers for mental health. X (formerly Twitter) provides an abundant amount of textual data from the public.[1] By considering tweets from users who have chosen to self-disclosed as having an MHC (Suhavi et al., 2022), we can analyse the differences in UED metrics across a diagnosis and the control population. We describe how utterance emotion dynamics (UED) metrics compare between different each of the seven diagnoses (ADHD, bipolar, depression, MDD, OCD, PPD, PTSD) and a control group; and for each MHC we explore the following research questions:

1. Does the average emotional state differ between the MHC and the control group?
2. Does emotional variability differ between the MHC and the control group?
3. Does the rate at which emotions reach peak emotional state (i.e., rise rate) differ between the MHC and the control group?
4. Does the rate at which emotions recover from peak emotional state back to steady state (i.e., recovery rate) differ between the MHC and the control group?

We explore each of the above research questions for three dimensions of emotion — valence, arousal, and dominance — further building on the findings in psychology which have traditionally focused only on valence. Our work provides baseline measures for UEDs across MHCs and insights into new linguistic biosocial markers for mental health. These findings are important for clinicians because they provide a broader context for overall well-being and can help contribute to the early detection, diagnosis, and management of MHCs.

---

[1]Since this work began before the name change, we use the terms *Twitter* and *tweets* in this paper.

## 2 Background

How people feel on average and the patterns of emotional change over time are well supported as a unique indicator of psychological well-being and psychopathology (Houben et al., 2015). Below we review related work on emotion dynamics and overall well-being in psychology and related fields.

### 2.1 Emotion Dynamics

Affective chronometry is a growing field of research that examines the temporal properties of emotions (i.e., emotion dynamics) and increasingly its relation to mental health and well-being have been studied (Kragel et al., 2022). Within this field, emotion reactivity is the threshold at which a response is elicited and the responses' amplitude and duration - it can be further deconstructed into the rise time to the peak response (i.e., rise rate), the maintenance time of the peak response, and the recovery time to baseline (i.e., recovery rate) (Davidson, 1998). Emotion variability is how variable an emotion is in terms of its intensity over time and in response to different stimuli. This variation can occur over multiple timescales (e.g., seconds, hours, days, weeks; Kragel et al., 2022). These emotion dynamic metrics are proposed to be predictive of affective trajectories over time and in distinguishing between affective disorders (Kragel et al., 2022). Also, emotion dynamics contribute to maladaptive emotion regulation patterns and poor psychological health (Houben et al., 2015). The meta-analysis by Houben et al. (2015) has indicated that the timescale of emotion dynamics does not moderate the relation between emotion dynamics and psychological well-being. Therefore, the relationship between psychological well-being and emotion dynamics occurs whether it is examined over seconds, days, months, and so on.

*Average Emotional State & Psychopathology:* Average or baseline emotional states are related to well-being and mental illnesses. Due to the maladaptive (i.e., dysfunctional) nature of psychopathology, those with mental illnesses tend to have more negative emotional baselines. For example, Heller et al. (2018) found that a higher average positive affect is associated with lower levels of depression but not anxiety, and a higher average negative affect was related to increased depression and anxiety. As well, those with post-traumatic stress disorder (PTSD) have reported lower average positive affect (Pugach et al., 2023).

*Emotion Reactivity & Psychopathology:* Research has found that individuals with psychopathologies tend to take longer to recover from differing emotional states (i.e., emotional resetting or recovery rate) than healthy individuals (Kragel et al., 2022). That is, difficulty moving between emotional states is associated with lower psychological well-being. Houben et al. (2015) also proposed that high emotional reactivity and slow recovery to baseline states is a maladaptive emotional pattern indicative of poor psychological well-being and psychopathology. In other words, people with poor psychological health may be highly reactive, emotionally volatile, and take a longer time to return to a baseline state.

*Emotion Variability & Psychopathology:* The Houben et al. (2015) meta-analysis findings also indicate that higher emotional variability is related to lower psychological well-being. In particular, variability was positively correlated with depression, anxiety, and other psychopathologies (e.g., bipolar, borderline personality disorder, etc.). This is supported by Heller et al. (2018) who found that higher positive and negative affect variability was associated with higher levels of depression and anxiety, these effects persisted for anxiety even after controlling for average positive affect. In contrast, variability was no longer associated with depression after controlling for average affect and the rate of recovery to baseline. This effect was attributed to anhedonia (the reduced ability to feel pleasure) which is a common symptom of depression that leads to reduced emotionality.

Overall, emotion dynamics research suggests that one's average emotional state, emotional variability, rise rate, and recovery rate may vary by their mental health. Preliminary research suggests that these metrics may also vary across different mental illnesses or psychopathologies. However, research on emotion dynamics within psychology and related fields has heavily relied on self-report measures and ecological momentary assessments (EMA). Affective self-report measures are subject to biases (Kragel et al., 2022) and thus carry certain limitations (i.e., social pressures to be perceived as happy). Additionally, data collection with these methods is time-intensive thus, sample size and study length are limited. Another window through which emotion dynamics can be inferred is through one's utterances (Hipson and Mohammad, 2021).

## 2.2 Utterance Emotion Dynamics

Early computational work on emotion change simply created emotion arcs using word–emotion association lexicons (Mohammad, 2011; Reagan et al., 2016). Teodorescu and Mohammad (2023) proposed a mechanism to evaluate automatically generated emotion arcs and showed that lexicon-based methods obtain near-perfect correlations with the true emotion arcs. The Utterance Emotion Dynamics (UED) framework uses various metrics inspired by psychology research to quantify patterns of emotion change from the emotion expressed in text (from the emotion arcs). Using a person's utterances allows researchers to analyse emotion dynamics since one's utterances can reasonably reflect one's thought process. UED allows for broader scale analyses across mediums (e.g., narratives, social media, etc.) and regions (e.g., cities, countries, etc.). UED metrics have been used to study the emotional trajectories of movie characters (Hipson and Mohammad, 2021) and to analyse emotional patterns across geographic regions through Twitter data (Vishnubhotla and Mohammad, 2022a). Seabrook et al. (2018) studied the association between depression severity and the emotion dynamics metric variability on Facebook and Twitter. The UED framework has also been applied to study developmental patterns of emotional change in poems written by children (Teodorescu et al., 2023).

This work explores the relationship between UEDs and mental health conditions. Also, unlike past work in emotion dynamics that has focused on valence (pleasure–displeasure or positive–negative dimension), this work also explores the arousal (active–sluggish) and dominance (in control–out of control, powerful–weak) dimensions. Together, valence, arousal, and dominance are considered the core dimensions of emotion (Russell, 2003).

## 3 Datasets

We use a recently compiled dataset—Twitter-STMHD (Suhavi et al., 2022). It comprises of tweets from 27,003 users who have self-reported as having a mental health diagnosis on Twitter. The diagnoses include: depression, major depressive disorder (MDD), post-partum depression (PPD), post-traumatic stress disorder (PTSD), attention-deficit/hyperactivity disorders (ADHD), anxiety, bipolar, and obsessive-compulsive disorder (OCD). We describe the dataset in Section 3.1, and our preprocessing steps in Section 3.3.

While our focus is on the relationship between emotion dynamics in tweets and MHCs, as a supplementary experiment, we also briefly explore the relationship between emotion dynamics in Reddit posts and depression.[2] Textual datasets associated with MHCs are not common, but it is beneficial to contextualize findings on tweets in comparison to findings on datasets from other modalities of communication. Due to the inherent differences in domains, dataset creation process, sample size, etc., we expect that there will be differences, however there may also be potential similarities in how UED relate to MHCs.

## 3.1 Twitter Dataset: STMHD

Suhavi et al. (2022) identified tweeters who self-disclosed an MHC diagnosis using carefully constructed regular expression patterns and manual verification. We summarize key details in the Appendix (Section A). The control group consists of users identified from a random sample of tweets (posted during roughly the same time period as the MHC tweets). These tweeters did not post any tweets that satisfied the MHC regex described above. Additionally, users who had any posts about mental health discourse were removed from the control group. Note that this does not guarantee that these users did not have an MHC diagnosis, but rather the set as a whole may have very few MHC tweeters. The number of users in the control group was selected to match the size of the depression dataset, which has the largest number of users.

For the finalized set of users, four years of tweets were collected for each user: two years before self-reporting a mental health diagnosis and two years after. For the control group, tweets were randomly sampled from between January 2017 and May 2021 (same date range as other MHC classes).

Users with less than 50 tweets collected were removed so as to allow for more generalizable conclusions to be drawn. Similarly, users with more than 5000 followers were removed so as not to include celebrities, or other organizations that use Twitter to discuss well-being.

## 3.2 Reddit Dataset: eRisk 2018

To further add to our findings, we also include the eRisk 2018 dataset (Losada et al., 2017, 2018) in our experiments. It consists of users who self-

---

[2]The available Reddit data only included information about depression; we hope future work will explore other MHCs.

| Dataset | Group | #People | Avg. #Posts/User |
|---|---|---|---|
| *Twitter* | | | |
| | MHC | 10,069 | 2,177.4 |
| | ADHD | 3,866 | 2,122.2 |
| | Bipolar | 721 | 3,193.3 |
| | Depression | 3,017 | 2,084.0 |
| | MDD | 133 | 2,402.9 |
| | OCD | 605 | 1,822.9 |
| | PPD | 105 | 1,671.4 |
| | PTSD | 1,622 | 1,944.9 |
| | Control | 4,097 | 1,613.6 |
| *Reddit* | | | |
| | Depression | 106 | 233.79 |
| | Control | 749 | 359.74 |

Table 1: The number of users in each mental health condition and the number of tweets per user in the pre-processed version of the Twitter-STMHD and Reddit eRisk datasets we use for experiments.

disclosed as having depression on Reddit (expressions were manually checked), and a control group (individuals were randomly sampled). Contrary to the Twitter-STMHD dataset where users in the control group were removed for discussing well-being topics, Losada et al. (2018) also considered users who discuss depression in the control group. The dataset includes several hundred posts per user, over approximately a 500-day period. We consider both the training set (which is from the eRisk 2017 task (Losada et al., 2017)) and the test set (from the eRisk 2018 task (Losada et al., 2018)).

## 3.3 Our Preprocessing

We further preprocessed both the Twitter-STMHD dataset and the eRisk dataset for our experiments (Section 4), as we are specifically interested in the unique patterns of UED for each disorder. Several users self-reported as being diagnosed with more than one disorder, referred to as *comorbidity*. We found a high comorbidity rate between users who self-reported as having anxiety and depression, as is also supported in the literature (Pollack, 2005; Gorman, 1996; Hirschfeld, 2001; Cummings et al., 2014). Therefore, we removed the anxiety class and only considered the depression class as it was the larger class between the two. We also performed the following preprocessing steps:

- We only consider users who self-reported as having one disorder. We removed 1272 users who had disclosed more than one diagnosis.
- We only consider tweets in English, removing other languages.
- We filtered out posts that contained URLs.
- We removed retweets (identified through tweets containing 'RT', 'rt').

- We computed the number of posts per user, and only considered users whose number of posts was within the interquartile range (between 25th and 75th percentile) for the diagnosis group they disclosed. This was to ensure that we are not including users that post very infrequently or very frequently.
- We removed punctuation and stop words.

Table 1 shows key details of the filtered Twitter-STMHD and Reddit e-risk datasets. We make our code for preprocessing the data publicly available.[3]

## 4 Experiments

To determine whether UED metrics from tweets can act as biosocial markers for psychopathology, we compare UED metrics for each MHC to the control group to see if they are statistically different. We compute the following UED metrics per user in each condition, following the *speaker UED* approach as described in Teodorescu et al. (2023). In this approach, UED metrics are computed per speaker by placing all their utterances in temporal order and computing UED on these ordered utterances. These metrics often rely on the characterization of the steady state or home base for emotions (regions of high probability). Hipson and Mohammad (2021) define the region pertaining to one standard deviation on either side of the mean as the *home base*. Previous work computed speaker UED metrics for characters in movie dialogues (Hipson and Mohammad, 2021), and for users on Twitter during the pandemic (Vishnubhotla and Mohammad, 2022a). We summarize the UED metrics below (also see Figure 1):

- **Average Emotional Intensity**: One's average emotion over time.
- **Emotional Variability**: How much and how often one's emotional state changes over time.
- **Rise Rate**: The rate at which one reaches peak emotional intensity from the home base (i.e., emotional reactivity).
- **Recovery Rate**: The rate at which one recovers from peak emotional intensity to home base, (i.e., emotional regulation).

For each user, we order their tweets by timestamp and used the Emotion Dynamics toolkit (Vishnubhotla and Mohammad, 2022b; Hipson and Mo-

[3]https://github.com/dteodore/EmotionArcs

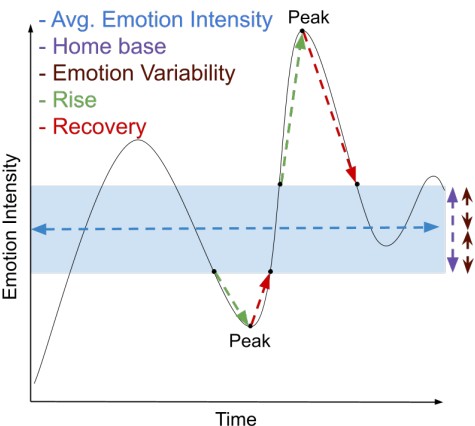

Figure 1: Utterance emotion dynamics metrics quantify patterns of emotional change over time.

hammad, 2021)[4] to compute UED metrics (average emotion, emotion variability, rise rate, and recovery rate). We performed analyses for valence, arousal, and dominance. For word-emotion association scores we use the NRC Valence, Arousal, and Dominance (VAD) lexicon (Mohammad, 2018).[5] It includes ∼20,000 common English words with scores ranging from −1 (lowest V/A/D) to 1 (highest V/A/D). Afterwards, we performed an ANOVA to test for significant differences between groups in UED metrics, and post-hoc analyses to determine which groups specifically had significant differences from the control group.

## 5   Results

To analyse potential differences across groups and the control group, an ANOVA statistical test can be conducted, however several assumptions must be met. The three primary assumptions are: the data for each independent variable are approximately normally distributed, the data are independent of each other, and the distributions have roughly the same variance (homoscedasticity). We can assume the mean is *normally distributed* according to the central limit theorem, due to the large sample size (law of large numbers). Since there are different tweeters in each MHC group we can assume the data are independent of each other. However, we note that people are largely not independent of each other e.g., having friends across groups, interacting with content from various mental health groups, etc. In our case, Levene's test indicated that the assumption for homogeneity of variance was violated

for all metrics and emotions (results in Appendix B.1). As such, we used Welch's ANOVA test which is an alternative to ANOVA when the equal variance assumption is not met. We conducted a 1-way Welch's ANOVA for each of the UED metrics (emotional average, emotional variability, rise-rate, and recovery rate) and emotions (valence, arousal, and dominance). We examined a total of $N = 14166$ users, see Table 1 for descriptives.

The first part of our analyses are Omnibus F-tests for significant differences between groups. This test cannot tell us which groups are different from each other, just rather that there is a difference among groups. For each combination of UED metrics and emotions (e.g., emotion variability for valence) there was a significant main effect which means we can conclude that at least one of the mental health diagnoses significantly differed. We show the degrees of freedom, F-statistic, p-value, and corrected effect size in Table 4 in the Appendix for valence, arousal, and dominance. The effect size tells us how meaningful the difference between groups is for each metric.[6] For example, 3.31% of the total variance in the emotional variability for valence is accounted for by diagnosis (small effect).

Next, we would like to know exactly which groups differed from the control group. In order to do this, we performed post hoc analyses for pairwise comparisons between groups for each metric across the three dimensions of emotions. We applied a Games-Howell correction since the assumption for homogeneity of variance was violated. In the following Sections we detail how the UED metrics compare across MHCs compared to the control group. In Table 2 we show the pairwise comparison results for which UED metrics and emotion combination significantly differed from the control group across diagnoses, and the direction of the difference. We also show results on the eRisk dataset in Table 2, and contrast our findings for depression between the Twitter and Reddit datasets.

We contextualize our results with previous findings in psychology and studies in NLP. We note that the relationship between patterns of emotion change and well-being for the dimensional emotions arousal and dominance are under-explored – our findings provide important benchmarks for these emotions and UED metrics more generally.

---

[4]https://github.com/Priya22/EmotionDynamics
[5]http://saifmohammad.com/WebPages/nrc-vad.html

[6]An effect size <0.01 is *very small*, between 0.01 to 0.06 is *small*, between 0.06 to 0.14 is *medium*, and greater than or equal to 0.14 is *large*.

| Dataset | MHC–Control | Average Emotion | | | Emotion Variability | | | Rise Rate | | | Recovery Rate | | |
|---|---|---|---|---|---|---|---|---|---|---|---|---|---|
| | | V | A | D | V | A | D | V | A | D | V | A | D |
| Twitter-STMHD | ADHD–control | ↓ | ↓ | ↓ | ↑ | ↑ | ↑ | – | – | ↑ | – | ↑ | ↑ |
| | Bipolar–control | – | ↓ | ↓ | ↑ | ↑ | ↑ | – | – | – | ↑ | – | – |
| | MDD–control | ↓ | – | ↓ | ↑ | ↑ | ↑ | ↑ | – | – | ↑ | ↑ | ↑ |
| | OCD–control | – | ↓ | ↓ | ↑ | ↑ | ↑ | – | – | ↑ | – | ↑ | ↑ |
| | PPD–control | – | ↓ | ↓ | – | ↑ | ↑ | – | – | – | – | – | – |
| | PTSD–control | ↓ | – | ↓ | ↑ | ↑ | ↑ | ↑ | ↑ | – | ↑ | ↑ | ↑ |
| | Depression–control | – | ↓ | ↓ | ↑ | ↑ | ↑ | ↑ | – | ↑ | ↑ | ↑ | ↑ |
| Reddit eRisk | Depression–control | – | – | ↓ | ↑ | – | ↑ | – | – | – | – | – | – |

Table 2: The difference in UED metrics between each MHC group and the control. A significant difference is indicated by an arrow; arrow direction indicates the direction of the difference. E.g., ↓ for ADHD–control and average emotion 'V' means that the ADHD group has significantly lower average valence than the control group.

## 5.1 How does the average emotion for an MHC compare to the control?

**Valence:** The average valence was significantly lower for the ADHD, MDD, and PTSD groups compared to the control group.

**Arousal:** The ADHD, depression, bipolar, PPD, and OCD groups showed significantly lower arousal compared to the control group.

**Dominance:** All MHC groups (ADHD, depression, bipolar, MDD, PPD, PTSD, OCD) showed significantly lower dominance compared to the control group. Additionally, the depression group in the eRisk dataset also had significantly lower dominance compared to the control group.

*Discussion:* Our findings align with results in psychology, and NLP studies looking at the average emotion intensity expressed in text. Valence was found to be lower in individuals with depression (of which MDD is a type of depression) through self-reports questionnaires (Heller et al., 2018; Silk et al., 2011) and on social media (Seabrook et al., 2018; De Choudhury et al., 2013, 2021). Further, work in psychology has found individuals with PTSD and ADHD have lower valence (Pugach et al., 2023; Stickley et al., 2018). It has also been shown that lower arousal and dominance in speech is associated with depression (Stasak et al., 2016; Osatuke et al., 2007; Gumus et al., 2023). While average emotion intensity is one of the more commonly explored measures, there is still relatively few works studying the relationships between arousal and dominance with mental health, compared to valence. Interestingly, dominance appears to differ for many MHCs (all studied here) from the control group, pointing towards an important indicator of well-being.

## 5.2 How does emotion variability for an MHC compare to the control?

**Valence:** Variability for valence was significantly higher for the ADHD, depression, bipolar, MDD, PTSD, and OCD groups compared to the control. PPD did not show differences from the control. The depression group in the eRisk dataset also showed significantly higher valence variability compared to the control group.

**Arousal:** All MHC groups (ADHD, depression, bipolar, MDD, PPD, PTSD, and OCD) showed significantly higher arousal variability.

**Dominance:** All MHC groups (ADHD, depression, bipolar, MDD, PPD, PTSD, OCD) had significantly higher dominance variability than the control group. The depression group in the eRisk dataset also had significantly higher dominance variability compared to the control group.

*Discussion:* In several studies in psychology, it has been shown that higher valence variability occurred for individuals with depression, PTSD (Houben et al., 2015; Heller et al., 2018) and is negatively correlated with overall well-being (Houben et al., 2015). Interestingly, Seabrook et al. (2018) found higher valence variability on Twitter indicated lower depression severity which contradicted their findings on Facebook. Kuppens et al. (2007) report that valence variability was negatively related to self-esteem and was positively related to neuroticism and depression. Overall, our results align with emotional variability having strong ties with well-being. Arousal and dominance variability appear to be *biosocial* markers across several MHCs, although minimally explored in the literature (Ranney et al. (2020) found higher affective arousal variability was associated with generalized anxiety disorder).

### 5.3 How does emotional rise rate for an MHC compare to the control?

**Valence:** Rise-rate for valence was significantly higher for the depression, MDD, and PTSD groups compared to the control group.
**Arousal:** PTSD was the only group which had statistically higher arousal rise rates than the control group.
**Dominance:** The ADHD, depression, and OCD groups had significantly higher rise rates than the control group.
*Discussion*: Rise-rate is analogous to emotional reactivity in psychology, and quickly moving to peak emotional states has been shown in individuals with maladaptive emotion patterns and lower psychological well-being (Houben et al., 2015). It is interesting to note that valence and dominance rise rates differed across MHC to the control, whereas not to the same extent for arousal.

### 5.4 How does emotional recovery rate for an MHC compare to the control?

**Valence:** Recovery rate for valence was significantly higher for the depression, bipolar, MDD, and PTSD groups compared to the control group.
**Arousal:** The ADHD, depression, MDD, PTSD, and OCD groups showed significantly higher arousal recovery rates than the control group.
**Dominance:** The ADHD, depression, MDD, PTSD and OCD groups showed significantly higher dominance recovery rates than the control group.
*Discussion*: Recovery rates can be thought of as a proxy of emotion regulation, and slower recovery from emotional events is associated with psychopathology and poor psychological well-being (Houben et al., 2015; Kragel et al., 2022). Our results, while pointing to higher recovery rates, indicate significant differences from the control group. This is an interesting result that can be further explored if found in other mediums such as Reddit.

We expected to see differences in which UED significantly different from the control between the Twitter and Reddit datasets due to the different mediums, collection process, etc. as mentioned in Section 3. However, we also saw commonalities in that average dominance was significantly lower, and valence and dominance variability were significantly higher for the depression group compared to the control. It appears that despite dataset differences, these *biosocial* markers are strong indicators for depression. Future work can explore further

whether rise and recovery rates act as indicators as well, perhaps by varying parameters involved in computing UED, such as the time-step granularity.

Overall, we notice that there are quite a few UED metrics across emotion dimensions which are significantly different for MHCs from the control group. Often also, that the direction of difference is similar for valence, arousal, and dominance. Work in psychology supports that often valence and dominance move in similar directions. For example, (Graziotin et al., 2015) found software engineers were more productive when in a good mood associated with higher valence (i.e., more positive) and higher dominance (i.e., more in control of one's feelings). Likewise, There may be several factors which influence valence and dominance to move in similar directions in the control group and MHCs.

## 6 Results: *Biosocial* Aspects

We now explore whether the UED metrics that were found to be significant indicators of certain MHCs, continue to remain discriminating even when accounting for certain social factors such as how popular a user's posts tend to be. Metadata for the number of likes per post is available in the dataset. So, we simply extracted this information and computed the average number of likes per tweeter. Most users had 0–4 likes per post in both the MHCs and the control group, up to an average of 8 likes per post. We compare the average UED per emotion dimension across the various bins of average likes. If a popularity measure does not impact UED, then we would expect to see consistent differences between the control group and an MHC across popularity bins. However, if a popularity measure does influence UED then the differences across MHCs and the control may disappear when controlling for popularity.

In Figure 2, we show the two UED metrics (e.g., average emotion intensity and emotional variability) for valence, across various bins of average likes per tweeter for each group. In the Appendix (Section D, E, and F), we show the remaining results for arousal and dominance. If a bin had less than ten users in it (e.g., average of 6–8 likes per post), then we did not plot a value for it. In each figure the control group is the blue line (no shapes). For MHCs which significantly differed from the control for an emotion and UED pair, we show the mean difference in Appendix G. If this difference between an MHC and the control remains fairly

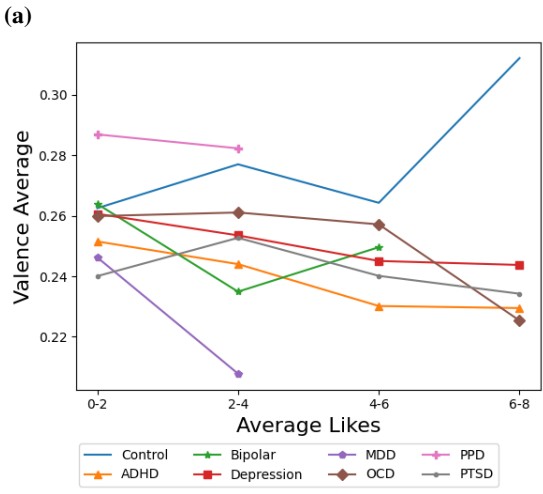 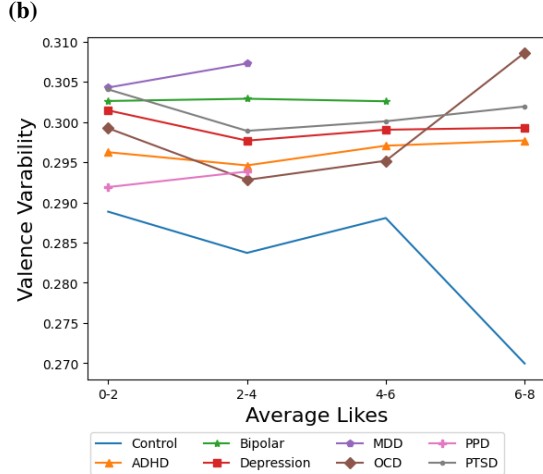

Figure 2: **Valence** average and variability across levels of user *popularity* on Twitter (average #likes on posts).

consistent across ranges of average likes, this UED is most likely not influenced by this social aspect.

*Average Valence:* In the previous section, we showed that average valence is markedly lower for ADHD, MDD, and PTSD. From Figure 2 (a), we observe that the average valence for these MHCs is markedly lower than control for various Average-Likes bins. (We also observe markedly greater differences when average likes is in the [6,8) range.)

*Valence Variability:* In the previous section, we showed that valence variability is markedly higher for all MHCs except PPD. From Figure 2 (b), we observe that the valence variability for these MHCs is markedly higher than control for various Average-Likes bins. (We also observe markedly lower variability for control when average likes is in the [6,8) range.)

*Valence Rise Rate:* In the previous section, we showed that valence rise rate is markedly higher for MDD, PTSD, and Depression. From Figure 3 (a) we observe that the valence rise rate for these MHCs is markedly higher than control for various Average-Likes bins. (We also observe markedly lower valence rise rate for control when average user likes is in the [6,8) range.)

*Valence Recovery Rate:* In the previous section, we showed that valence recovery rate is markedly higher for Bipolar, MDD, PTSD, and Depression. From Figure 3 (b) we observe that the valence recovery rate for these MHCs is markedly higher than control for various Average-Likes bins. (We also observe markedly lower recovery rate for control when average user likes is in the [6,8) range.)

Thus, overall, we observe that for the valence UEDs which showed significant differences across MHCs and control, these differences do not disappear when accounting for popularity measures such as the average number of likes a user's posts get. Similar trends overall were found for arousal and dominance (Appendix E and F). UED for all three emotion dimensions appear to be robust indicators for various mental health conditions, even when users have some varying social characteristics.

## 7  Conclusion

We showed for the first time that there are significant relationships between patterns of emotion change in text written by individuals with a self-disclosed MHC compared to a control group. Specifically, we found significant differences in four utterance emotion dynamics (UED) metrics (average emotion, emotion variability, rise rate, and recovery rate) across three emotion dimensions (valence, arousal, and dominance) for 7 MHCs. Our findings provide important contextual information of overall well-being and supporting indicators (in addition to other assessments) to clinicians for diagnosis detection and management.

Looking ahead, we plan to explore UED metrics across different textual genres, regions, languages, and demographics (such as socioeconomic status), in collaboration with clinicians. Through such a partnership, UED metrics could be studied in the context of clinical information as well. Lastly, exploring new UED metrics that correlate with psychopathology is a promising direction for future work.

## Limitations

In this study, we used NLP techniques to compare UED across different psychopathologies and a control group. It is important to be cautious of interpreting these results due to natural limitations within the dataset. Due to the high rates of comorbidity with mental health disorders (Druss and Walker, 2011) examining users who only disclosed one diagnosis may not be representative of the population. Furthermore, it is also possible that the dataset included users with more than one disorder but only disclosed one (e.g., a user may have been diagnosed with depression and ADHD but only tweeted "diagnosed with ADHD" or vice versa). Self-disclosure of diagnoses may also be inaccurate due to reasons such as impression management (Leary, 2001) or social desirability (Latkin et al., 2017) where users may disclose having a diagnosis without having received a formal diagnosis. Alternatively, there may have been users included in the control group who have a formal diagnosis of one or more mental health disorders but did not disclose this on Twitter. Overall, despite the dataset creators' best efforts to collect data accordingly, the users in each group may not be representative of the mental disorders. Future research could replicate this study using a sample of users with confirmed formal diagnoses.

## Ethics Statement

Our research interest is to study emotions at an aggregate/group level. This has applications in emotional development psychology and in public health (e.g., overall well-being and mental health). However, emotions are complex, private, and central to an individual's experience. Additionally, each individual expresses emotion differently through language, which results in large amounts of variation. Therefore, several ethical considerations should be accounted for when performing any textual analysis of emotions (Mohammad, 2022, 2023). The ones we would particularly like to highlight are listed below:

- Our work on studying emotion word usage should not be construed as detecting how people feel; rather, we draw inferences on the emotions that are conveyed by users via the language that they use.
- The language used in an utterance may convey information about the emotional state (or perceived emotional state) of the speaker, listener, or someone mentioned in the utterance. However, it is not sufficient for accurately determining any of their momentary emotional states. Deciphering the true momentary emotional state of an individual requires extra-linguistic context and world knowledge. Even then, one can be easily mistaken.
- The inferences we draw in this paper are based on aggregate trends across large populations. We do not draw conclusions about specific individuals or momentary emotional states.

## Acknowledgements

Many thanks to Krishnapriya Vishnubhotla for the Emotion Dynamics codebase, which set the groundwork for computing emotion dynamics, and for insightful discussions. This research was supported by NSERC, SSHRC, Digital Research Alliance of Canada (alliancecan.ca), Alberta Innovates, DeepMind, and CIFAR. Alona Fyshe holds a Canada CIFAR AI Chair. This research project is funded by the Bavarian Research Institute for Digital Transformation (bidt), an institute of the Bavarian Academy of Sciences and Humanities. The author is responsible for the content of this publication.

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

# APPENDIX

## A  Twitter-STMHD Dataset

Suhavi et al. (2022) created a regular expression pattern to identify posts which contained a self-disclosure of a diagnosis and the diagnosis name (using a lexicon of common synonyms, abbreviations, etc.) such as 'diagnosed with X'. They collected a large set of tweets using the regex. This resulted in a preliminary dataset of users with potential MHC diagnoses. To handle false positives (e.g., 'my family member has been diagnosed with X', or 'I was not diagnosed with X'), the dataset was split into two non-overlapping parts, one of which was manually annotated, and the other using an updated and high-precision regex. In the part that was annotated by hand, each tweet was annotated by two members of the team. A user was only included in the dataset if both annotations were positive as self-disclosing for a particular class. A licensed clinical psychologist found the 500-tweet sample to be 99.2% accurate. The manual annotations were used to refine the regular expressions and diagnosis name lexicon. This updated search pattern was applied to the other dataset split. To verify the quality of the updated regex, the authors applied it to the manually annotated dataset split. When considering the manual annotations as correct, the regex was found to be 94% accurate.

## B  Statistical Results

### B.1  Levene's Test

Levene's test indicated that the assumption for homogeneity of variance was violated for the effect of diagnosis on all UED metrics (emotional average, emotional variability, rise rate, and recovery rate) across all three dimensional emotions (valence, arousal, and dominance). We show the results in Table 3.

### B.2  Welch's ANOVA

In Table 4 we show the results for Welch's ANOVA for valence, arousal, and dominance.

## C  Data Descriptives for Distributions of Social Aspects

In Table 5 we show some key details about the distribution of the *popularity* aspect on Twitter: average likes per Tweeter.

## D  UED Across Average Likes per Tweeter: Valence

In Figure 3, we show two UED metrics (e.g., rise rate and recovery rate) for valence, across various bins of average likes received for each group.

## E  UED Across Average Likes per Tweeter: Arousal

In Figure 4, we show the four UED metrics (e.g., average emotion intensity, emotional variability, rise rate, and recovery rate) for arousal, across various bins of average likes received for each group.
*Average Arousal:* In Section 5, we showed that average arousal is markedly lower for ADHD, Bipolar, OCD, PPD, and Depression. From Figure 4 (a), we observe that the average arousal for these MHCs is markedly lower than control for various Average-Likes bins. (We also observe slightly smaller differences for Depression and OCD when average likes is in the [6,8) range, perhaps pointing to the potential for average arousal to be influenced when a post is considered quite *popular*.)

*Arousal Variability:* In Section 5, we showed that arousal variability is markedly higher for all MHCs. From Figure 4 (b), we observe that the arousal variability for these MHCs is markedly higher than control for various Average-Likes bins. (We also observe markedly lower variability for control when average likes is in the [6,8) range.)

| Emotion | UED Metric | df1 | df2 | F-statistic | P-value |
|---------|-----------|-----|-----|-------------|---------|
| Valence | average emotion | 7 | 14158 | 60.50 | $p<.001$ |
| | emotional variability | 7 | 14158 | 66.33 | $p<.001$ |
| | rise rate | 7 | 14156 | 77.35 | $p<.001$ |
| | recovery rate | 7 | 14156 | 72.58 | $p<.001$ |
| Arousal | average emotion | 7 | 14158 | 37.60 | $p<.001$ |
| | emotional variability | 7 | 14158 | 22.38 | $p<.001$ |
| | rise rate | 7 | 14155 | 34.76 | $p<.001$ |
| | recovery rate | 7 | 14155 | 41.28 | $p<.001$ |
| Dominance | average emotion | 7 | 14158 | 61.86 | $p<.001$ |
| | emotional variability | 7 | 14158 | 72.21 | $p<.001$ |
| | rise rate | 7 | 14150 | 37.69 | $p<.001$ |
| | recovery rate | 7 | 14154 | 35.64 | $p<.001$ |

Table 3: The degrees of freedom, F-statistic, and p-value in Levene's test of Homogeneity of Variances for each UED metric and emotion.

| Emotion | UED Metric | df1 | df2 | F-statistic | P-value | Effect Size ($est\ \omega^2$) |
|---------|-----------|-----|-----|-------------|---------|------------------------------|
| Valence | average emotion | 7 | 1021.65 | 14.79 | $p<.001$ | 0.0068 |
| | emotional variability | 7 | 1021.20 | 70.30 | $p<.001$ | 0.0331 |
| | rise rate | 7 | 1026.32 | 9.93 | $p<.001$ | 0.0044 |
| | recovery rate | 7 | 1023.62 | 8.86 | $p<.001$ | 0.0039 |
| Arousal | average emotion | 7 | 1024.41 | 33.24 | $p<.001$ | 0.0157 |
| | emotional variability | 7 | 1029.77 | 66.23 | $p<.001$ | 0.0312 |
| | rise rate | 7 | 1025.85 | 2.84 | $p=.006$ | 0.0009 |
| | recovery rate | 7 | 1026.95 | 5.19 | $p<.001$ | 0.0021 |
| Dominance | average emotion | 7 | 1020.10 | 56.69 | $p<.001$ | 0.0268 |
| | emotional variability | 7 | 1023.12 | 40.50 | $p<.001$ | 0.0191 |
| | rise rate | 7 | 1025.35 | 6.31 | $p<.001$ | 0.0026 |
| | recovery rate | 7 | 1022.99 | 9.94 | $p<.001$ | 0.0044 |

Table 4: The degrees of freedom (for the numerator and denominator), F-statistic, p-value, and effect size in Welch's ANOVA test for differences between groups, for each UED metric and emotion (valence, arousal, and dominance).

| Popularity Aspect | Average | Std. Dev. | 25th Percentile | Median | 75th Percentile |
|-------------------|---------|-----------|-----------------|--------|-----------------|
| Avg. Likes per Tweeter | 2.06 | 8.33 | 0.67 | 1.20 | 2.11 |

Table 5: Key statistics of the distribution of the *popularity* aspect: Avg. Likes per Tweeter.

*Arousal Rise Rate:* In Section 5, we showed that arousal rise rate is markedly higher for PTSD. From Figure 4 (c), we observe that the arousal rise rate for PTSD does cross the control line at bin 3 with [4-6] likes. This points to the potential for arousal rise rate to be slightly influenced by a popularity aspect such as the average number of likes received. (We also observe markedly lower arousal rise rate for control when average user likes is in the [6,8) range.)

*Arousal Recovery Rate:* In Section 5, we showed that arousal recovery rate is markedly higher for ADHD, MDD, OCD, PTSD, and Depression. From Figure 4 (d), we observe that the arousal recovery rate for these MHCs is markedly higher than control for various Average-Likes bins (slightly less for ADHD). (We also observe markedly lower arousal recovery rate for control when average user likes is in the [6,8) range.)

Thus, overall, we observe that for the arousal UEDs which showed significant differences across MHCs and control, these differences still do appear (although slightly less for arousal rise and recovery rates) when accounting for popularity measures such as the average number of likes a user's posts get.

## F  UED Across Average Likes per Tweeter: Dominance

In Figure 5, we show the four UED metrics (e.g., average emotion intensity, emotional variability, rise rate, and recovery rate) for dominance, across various bins of average likes received for each group.
*Average Dominance:* In Section 5, we showed that average dominance is markedly lower for all MHCs. From Figure 5 (a), we observe that the average dominance for these MHCs is markedly lower than control for various Average-Likes bins.

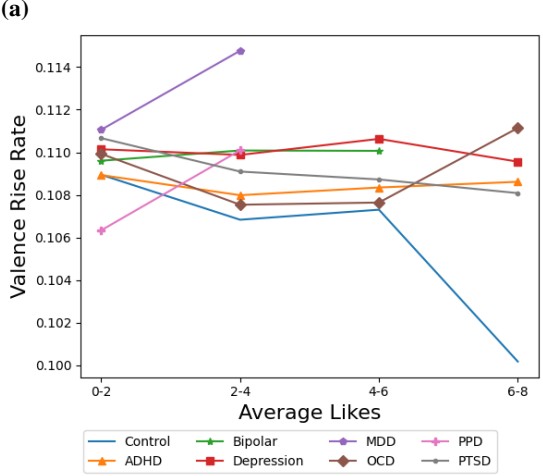
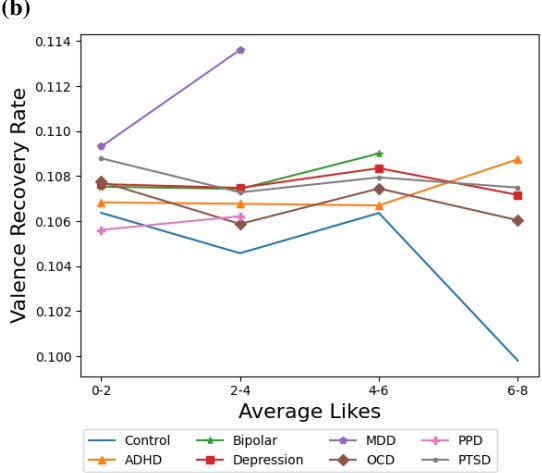

Figure 3: **Valence** rise and recovery rates across levels of user *popularity* on Twitter (average #likes on posts).

*Dominance Variability:* In Section 5, we showed that dominance variability is markedly higher for all MHCs. From Figure 5 (b), we observe that the dominance variability for these MHCs is markedly higher than control for various Average-Likes bins. (We also observe markedly lower variability for control when average likes is in the [6,8) range.)

*Dominance Rise Rate:* In Section 5, we showed that dominance rise rate is markedly higher for ADHD, OCD and Depression. From Figure 5 (c), we observe that the dominance rise rate for these MHCs is markedly higher than control for various Average-Likes bins. (We also observe markedly lower dominance rise rate for control when average user likes is in the [6,8) range.)

*Dominance Recovery Rate:* In Section 5, we showed that dominance recovery rate is markedly higher for ADHD, MDD, OCD, PTSD, and Depression. From Figure 5 (d), we observe that the dominance recovery rate for these MHCs is higher than control for various Average-Likes bins although not for ADHD, PTSD which intersect the control line at bin 3 with [4-6) likes. We also notice that the difference dominance recovery rate for OCD and control is less at bin 3 with [4-6) likes. This points to the potential that dominance recovery rate may be influenced by a popularity measure such as the average number of likes received. (We also observe markedly lower dominance recovery rate for control when average user likes is in the [6,8) range.)

Thus, overall, we observe that for the dominance UEDs which showed significant differences across MHCs and control, these differences still do largely

appear (although less for dominance recovery rates) when accounting for popularity measures such as the average number of likes a user's posts get.

# G UED Mean Differences

Table 6 and Table 7 show the mean difference in UEDs between the control group and an MHC if the difference was statistically significant in the pairwise comparison shown in Section 5.

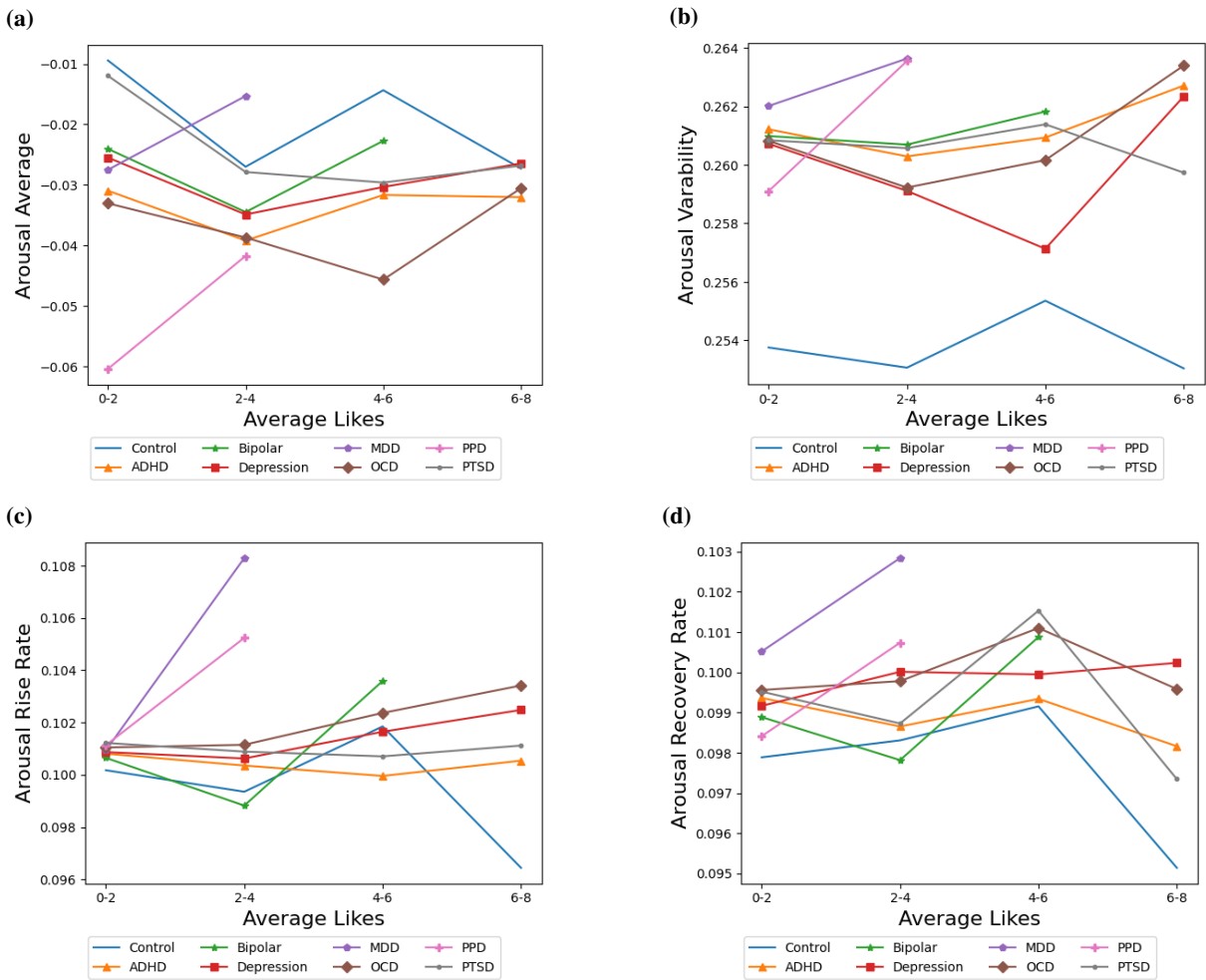

Figure 4: UED metrics for **arousal** across various levels of user *popularity* on Twitter (average #likes on posts).

| | | Average Emotion | | | Emotion Variability | | |
|---|---|---|---|---|---|---|---|
| **Dataset** | **MHC–Control** | V | A | D | V | A | D |
| Twitter-STMHD | ADHD–control | -0.018 | -0.021 | -0.030 | 0.008 | 0.007 | 0.006 |
| | Bipolar–control | – | -0.013 | -0.029 | 0.015 | 0.007 | 0.008 |
| | MDD–control | -0.027 | – | -0.059 | 0.017 | 0.009 | 0.009 |
| | OCD–control | – | -0.023 | -0.033 | 0.010 | 0.007 | 0.005 |
| | PPD–control | – | -0.046 | -0.051 | – | 0.006 | 0.006 |
| | PTSD–control | -0.023 | – | -0.018 | 0.015 | 0.007 | 0.006 |
| | Depression–control | – | -0.015 | -0.040 | 0.013 | 0.007 | 0.006 |
| Reddit eRisk | Depression–control | – | – | -0.041 | 0.014 | – | 0.006 |

Table 6: The mean difference in UED metrics between each MHC group and the control if there was a significant difference between groups. E.g., $-0.018$ for ADHD–control and average emotion 'V' means that the ADHD group has significantly lower average valence by $0.018$ than the control group.

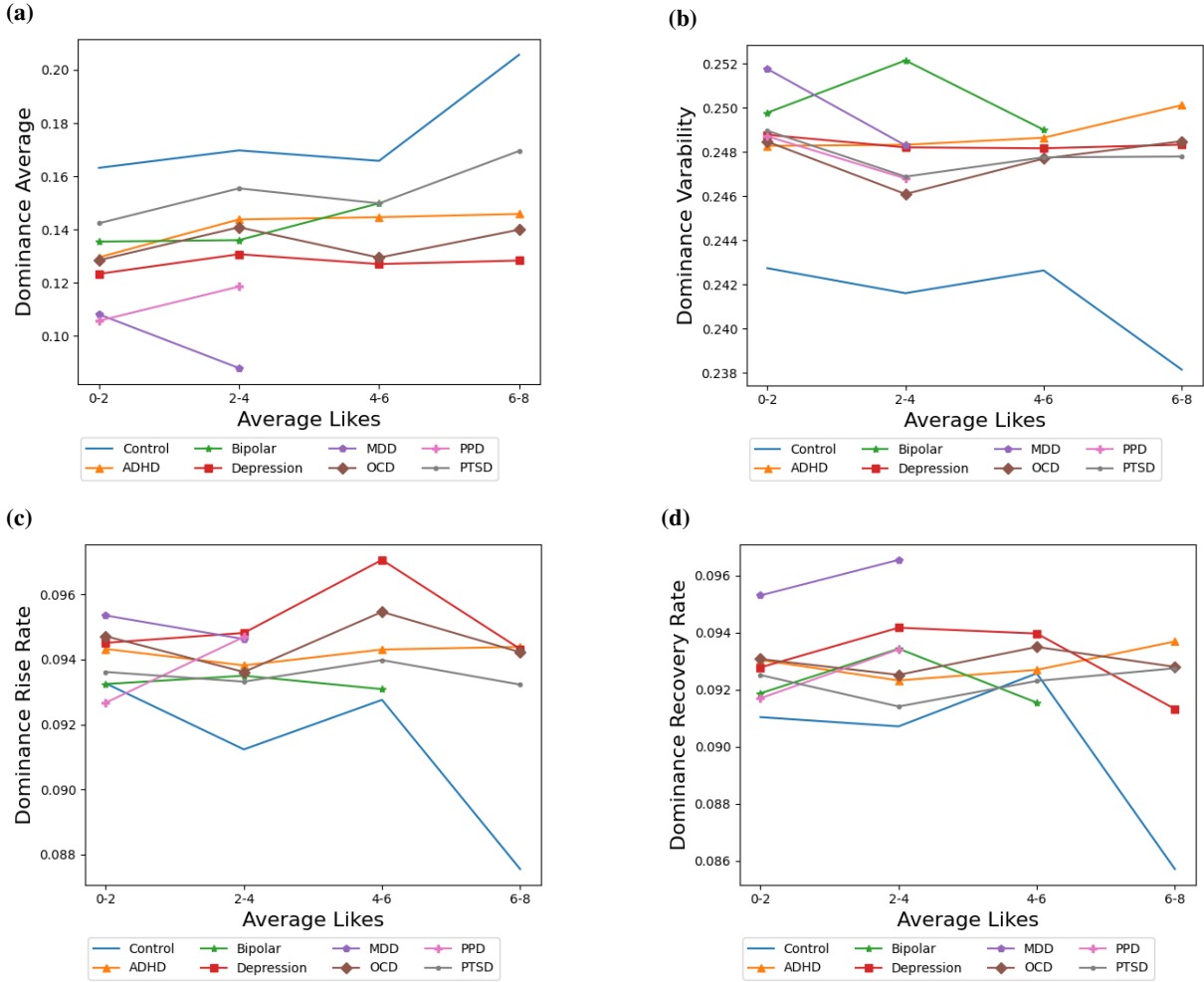

Figure 5: UED metrics for **dominance** across various levels of user *popularity* on Twitter (average #likes on posts).

| | | Rise Rate | | | Recovery Rate | | |
|---|---|---|---|---|---|---|---|
| **Dataset** | **MHC–Control** | V | A | D | V | A | D |
| Twitter-STMHD | ADHD–control | – | – | 0.001 | – | 0.001 | 0.002 |
| | Bipolar–control | – | – | – | 0.002 | – | – |
| | MDD–control | 0.004 | – | – | 0.004 | 0.003 | 0.005 |
| | OCD–control | – | – | 0.002 | – | 0.002 | 0.002 |
| | PPD–control | – | – | – | – | – | – |
| | PTSD–control | 0.002 | 0.001 | – | 0.002 | 0.001 | 0.001 |
| | Depression–control | 0.002 | – | 0.002 | 0.002 | 0.001 | 0.002 |
| Reddit eRisk | Depression–control | – | – | – | – | – | – |

Table 7: The mean difference in UED metrics between each MHC group and the control if there was a significant difference between groups. E.g., 0.004 for MDD–control and rise rate 'V' means that the MDD group has significantly higher rise rate for valence by 0.004 than the control group.