# OpenReview forum: "Language and Mental Health: Measures of Emotion Dynamics from Text as Linguistic Biosocial Markers"
_EMNLP/2023/Conference — EMNLP 2023 Main_

### Official Review · Reviewer_Vx9h · 2023-08-05

**Soundness:** 4

**Excitement:**

4: Strong: This paper deepens the understanding of some phenomenon or lowers the barriers to an existing research direction.

**Missing References:**

The 2022 CLPsych shared task seems super relevant:

Tsakalidis, A., Chim, J., Bilal, I. M., Zirikly, A., Atzil-Slonim, D., Nanni, F., ... & Liakata, M. (2022, July). Overview of the CLPsych 2022 shared task: Capturing moments of change in longitudinal user posts. In Proceedings of the Eighth Workshop on Computational Linguistics and Clinical Psychology (pp. 184-198).

This also seems relevant, as they show language predicting mental health diagnoses is better when the language is measured closer in time to the diagnosis:

Eichstaedt, J. C., Smith, R. J., Merchant, R. M., Ungar, L. H., Crutchley, P., Preoţiuc-Pietro, D., ... & Schwartz, H. A. (2018). Facebook language predicts depression in medical records. Proceedings of the National Academy of Sciences, 115(44), 11203-11208.

Also, there is some work that shows that emotion and mental health can be tracked across *populations* temporally. See Hannah Metzler & David Garcia for great work in this area as well as:

Mangalik, S., Eichstaedt, J. C., Giorgi, S., Mun, J., Ahmed, F., Gill, G., ... & Schwartz, H. A. (2023). Robust language-based mental health assessments in time and space through social media. arXiv preprint arXiv:2302.12952.

**Paper Topic And Main Contributions:**

The paper attempts to measure emotion dynamics (average emotion, emotion variability, rise rate, and recovery rate) using a data set of Twitter users who have self disclosed various mental health conditions (MHCs). The authors then show that emotion dynamics (where emotions are defined as valence, arousal, and dominance) tend to be different when comparing Twitter users with MHCs to those who do not.

**Questions For The Authors:**

* Did you apply a Bonferroni correction (for example) to adjust for your large number of comparisons?

* For Table 3, what does "across groups" mean? I thought groups meant the MHCs.

* Footnote 2 ("An effect size..."), where did you get these levels?

**Reasons To Accept:**

The authors address an important question. Humans (and their language) are temporally dynamic (as well as many other things, e.g. hierarchical) and most NLP methods to measure emotion do not take this into consideration. This is especially important as it relates to mental health.

**Reasons To Reject:**

The paper doesn't feel fully fleshed out in terms of its implications. We have 4 metrics, 3 emotions, and 7 MHCs, which is a fairly large space. Do we have any reason to suspect that there will be differential patterns to the results which depend on metric/emotion/MHC? For example, should rise rate for dominance matter for depression?

If "yes", then I think there are some issues. First, methodologically, the main weakness of the paper is in the VAD lexicon. I'm not convinced there are any meaningful differences between V, A, and D. For example, looking at the columns in Table 4 or Tables 3/5/6. I would imagine the correlations between the V, A, and D scores are very high. This is most likely a problem with the model, but, again, if your claim is that are differential patterns to these dynamics, then you need to show that this model is measuring three different things.

Second, your results need to be unpacked more. How does this all relate back to psychopathology? Are there any takeaways other than "we should measure dynamics"? A table highlighting *expected* associations (based on literature or a preregistration) would be super helpful.

If "no", then it's unclear to me that the 4 research questions are actually separate questions. The authors simply restate the same question but use a different UED. It seems like there is one central question: do dynamics differ between controls and MHCs? To be fair, the authors claim "this work provides important early evidence", so I don't think the authors are actually over-claiming anything, but I do think the paper could be reframed.

**Reproducibility:**

4: Could mostly reproduce the results, but there may be some variation because of sample variance or minor variations in their interpretation of the protocol or method.

**Reviewer Confidence:**

3: Pretty sure, but there's a chance I missed something. Although I have a good feel for this area in general, I did not carefully check the paper's details, e.g., the math, experimental design, or novelty.

**Typos Grammar Style And Presentation Improvements:**

Figure 1 (a) and (b) have the same caption.

Table 4: You are already using "*" to mark significance in Table 3, so please don't use that symbol for something else.

Table 4: I think this table could be more readable if you simply used V, A, and D instead of *, +, and a triangle.

Line 634: "valence, arousal, and, dominance" should have one less comma

---

> ### Author Rebuttal · Authors · 2023-08-28
>
> Dear reviewer,
>
> Thank you very much for your feedback and questions. We address them below.
>
> Re VAD and V-A, A-D, V-D correlations:
>
> We appreciate you bringing these points up. The VAD Lexicon is a widely used resource in NLP, Psychology, and many other disciplines. It is especially popular in works exploring mental health and language where interpretability is of utmost importance. It has close to 500 citations since the 5 years of its publication at ACL 2018 main conference, including many in the CL Psych literature.
>
> Re I would imagine the correlations between the V, A, and D scores are very high.
>
> This is not the case. The inter-dimension correlations listed in the 2018 ACL paper are:
> V–A: -0.26
> A–D: -0.30
> D–V: 0.48
> These indicate a weak inverse correlation for V–A and A–D, and a moderate correlation for D–V. (These inter-dimension correlations are similar to what Warriner et al. 2014 found in their lexicon.) This demonstrably shows the V, A, and D are measuring different things.
>
> Dominance being the dimension of helplessness – having a feeling of control,
>
> And
>
> Arousal being the dimension of calmness/sluggishness – excitement/anxiety
>
> have direct relevance to mental health conditions (the feelings of helplessness, dullness, and anxiety are known correlates with conditions such as depression, anxiety disorder, etc.)
>
> Johnson, S. L., Leedom, L. J., & Muhtadie, L. (2012). The dominance behavioral system and psychopathology: evidence from self-report, observational, and biological studies. Psychological bulletin, 138(4), 692–743. https://doi.org/10.1037/a0027503
>
> Mehrabian, A. (1997). Comparison of the PAD and PANAS as models for describing emotions and for differentiating anxiety from depression. Journal of Psychopathology and Behavioral Assessment, 19(4), 331–357. https://doi.org/10.1007/BF02229025
>
> Mehrabian, A. (1995). Distinguishing depression and trait anxiety in terms of basic dimensions of temperament. Imagination, Cognition and Personality, 15(2), 133–143. https://doi.org/10.2190/JB3J-LL1E-GYGY-D0RJ
>
> Work in Psychology has largely only explored valence – notably because it has long been shown the most important emotional signal, and most easy to obtain through self-reports (compared to other dimensions of emotions). Since the UED approach and the VAD lexicon allow exploration of arousal and dominance and because these two emotional dimensions are highly relevant to mental health conditions, we explore them here in this work.
>
> Unpacking results:
>
> We will add in a table highlighting the expected associations based on literature in psychology to help better contextualize our findings.
>
> We are interested in whether UED is a meaningful indicator of mental health and well-being. While that is our overarching research question, we explore this through the various metrics in the UED framework (e.g., average emotion intensity, variability, rise rate and recovery rate). Analyzing the relationship for each UED marker between MHCs and the control then become subquestions, allowing us to answer our overarching question of if UED are biosocial markers for mental health.
>
> Q: Did you apply a Bonferroni correction (for example) to adjust for your large number of comparisons?
>
> A: Yes, we applied the Games Howell correction for multiple comparisons. We will state it explicitly in the paper.
>
> Q: For Table 3, what does "across groups" mean? I thought groups meant the MHCs.
>
> A: By “across groups”, this is referring to each of the MHCs. We will update this caption to make it more clear as the following:
>
> “Valence: The mean UED metrics for each MHC. Those with an asterisk were significantly different from the control group.”
>
> Q: Footnote 2 ("An effect size..."), where did you get these levels?
>
> A: These levels are from Fields, 2013. Andy Fields is a well-known and respected statistician. However there are also guidelines by Cohen, 1992. As selecting effect size guidelines is largely dependent on context and customs in the field, we select Fields’ guidelines since we are performing exploratory analyses and predict that UED will be more difficult to predict.
>
> Field, Andy. 2013. Discovering Statistics Using IBM SPSS Statistics. sage.
> Cohen, Jacob. 1992. “A Power Primer.” Psychological Bulletin 112 (1): 155.
>
> Thank you for those references. They are relevant as you are mentioning and we have added them in. Also, we’ve addressed the typos & comments in the paper.
>
> We thank you for your thoughtful review and engaging with our work. We see the time and thought you have put into this and very much appreciate it.

---

### Official Review · Reviewer_9kEM · 2023-08-05

**Typos Grammar Style And Presentation Improvements:** Figure 1 - the title is the same for …
**Soundness:** 3

**Excitement:**

4: Strong: This paper deepens the understanding of some phenomenon or lowers the barriers to an existing research direction.

**Paper Topic And Main Contributions:**

The authors demonstrate correlation between patterns of emotion change learnt from language and mental health disorders on an existing Twitter dataset with a focus to examine if UED can be considered a biosocial marker for mental health.

**Questions For The Authors:**

Questions:

A Line 621-624 - Emotions also vary due to one's age, gender and culture. E.g. Young adults are more likely to express arousal. Pride in Western society has higher positive valence and arousal compared to Japan. It is thus important to also control for age, gender and location of users. There are ML based methods to predict user demographics which authors may explore.

See Batja Mesquita, Nico H Frijda, and Klaus R Scherer.
1997. Culture and emotion. Handbook of crosscultural psychology, 2:255–297

Emi Furukawa, June Tangney, and Fumiko Higashibara.
2012. Cross-cultural continuities and discontinuities
in shame, guilt, and pride: A study of children residing in japan, korea and the usa. Self and Identity,
11(1):90–113.

B. In Sec 5.4, the high recovery rate for MHC group and similar arousal for emotional rise-rate - raise the question if there is an underlying platform effect (Twitter) for emotion variability. The authors might consider analysis on another social media platform.


**Reasons To Accept:**

Emotion variability as a marker of mental health disorders, is intuitive yet understudied. The authors provide a detailed analysis highlighting its potential for detecting MHC which could be of interest to research community working on similar problems.

**Reasons To Reject:**

See Questions for the authors.

**Reproducibility:**

3: Could reproduce the results with some difficulty. The settings of parameters are underspecified or subjectively determined; the training/evaluation data are not widely available.

**Reviewer Confidence:**

3: Pretty sure, but there's a chance I missed something. Although I have a good feel for this area in general, I did not carefully check the paper's details, e.g., the math, experimental design, or novelty.

---

> ### Author Rebuttal · Authors · 2023-08-28
>
> Dear reviewer,
>
> Thank you very much for your feedback and questions. We appreciate your time in reviewing our work. We address them below.
>
> We fully agree with you that more can be done regarding the social aspects of biomarkers. We would like to clarify that the primary goal of this work is to assess the utterance emotion dynamics in language used by various populations associated with mental health conditions. That said, regardless of the extent to which social aspects are experimented with in a work, we think it is always important to refer to such linguistic markers as biosocial. Not treating them so has often, in the past, led to over generalized claims and one-size fit all solutions, at the expense of marginalized communities. Thus, our introduction situates the description of UED markers firmly as bio social. We also point out below some challenges with addressing social aspects responsibly which we will add in the revised version. Further, we did not want to force-fit more aspects when there wouldn't be adequate space to describe them. We hope for more careful and thoughtful work in this space in the future.
>
> Controlling for age, gender, culture:
>
> In our work, we control for popularity measures on social media as one such aspect. Some other aspects worth exploring include demographics such as age, gender, region, and culture as you are mentioning. We could control for gender, however this has become a sensitive topic as of late. Twitter’s current policy states:
>
> “You should be careful about using Twitter data to derive or infer potentially sensitive characteristics about Twitter users. Never derive or infer, or store derived or inferred, information about a Twitter user’s: Health (including pregnancy), Negative financial status or condition, Political affiliation or beliefs, **Racial or ethnic origin**, or beliefs, Sex life or sexual orientation, Trade union membership, Alleged or actual commission of a crime.”
>
> Emotion Variability on Twitter:
>
> Yes, exactly as you are pointing out. Variability could have an impact on some measures. In fact, looking into this further after the submission deadline we found what is referred to as the “inertia-instability paradox” in psychology (Koval et al., 2013; Bos et al., 2019). When controlling for variability it appears that these two emotion dynamics metrics are no longer associated with depressive symptoms (Koval et al., 2013; Bos et al., 2019). We will add in analyses controlling for variability to see what role variability plays in UED.
>
> Koval, Peter et al. “Affect dynamics in relation to depressive symptoms: variable, unstable or inert?.” Emotion (Washington, D.C.) vol. 13,6 (2013): 1132-41. doi:10.1037/a0033579
>
> Bos, Elisabeth H et al. “Affective variability in depression: Revisiting the inertia-instability paradox.” British journal of psychology (London, England : 1953) vol. 110,4 (2019): 814-827. doi:10.1111/bjop.12372
>
> We thank you for your thoughtful review and engaging with our work. We see the time and thought you have put into this and very much appreciate it.

---

### Official Review · Reviewer_Gsi5 · 2023-08-05

**Soundness:** 3

**Excitement:**

4: Strong: This paper deepens the understanding of some phenomenon or lowers the barriers to an existing research direction.

**Missing References:**

There are at least a couple of other studies which have discussed the idea of using emotion variability over time as a criteria for detecting mental health disorders, even if not using the same metrics:

Uban, Ana-Sabina, Berta Chulvi, and Paolo Rosso. "An emotion and cognitive based analysis of mental health disorders from social media data." Future Generation Computer Systems 124 (2021): 480-494.




**Paper Topic And Main Contributions:**

This paper proposes a study on the relationship between emotion dynamics, as reflected in texts posted on social media, and a self-stated diagnosis of a mental disorder, including several different mental disorders.
The novelty is in the focus on emotion variability (aside from the usual emotional valence) in relation to mental health, measured with different metrics. The paper also introduces the term "biosocial marker", referring to how emotional dynamics can be useful as a diagnostic factor for mental health, but only seen in a social and cultural context (manifestations can be different depending on culture or demographics). The introduction is interesting and well developed.

The study uses an existing Twitter dataset. The methods are relatively simple in terms of natural language processing (measures of different emotion metrics based on lexicons), but very robust in terms of statistical analysis of the data.
Some of the results are novel, especially given the new metrics used compared to the usual studies; the inclusion of several different disorders is also a strength.

While the authors insist on the "biosocial marker" motivation in the introduction and explain it well, their methodology and discussion does little to accomplish the goal of studying the problem from this perspective, in my opinion. The contextualization of the relation between emotion dynamics and mental disorder diagnosis wrt culture or demographics is very brief. Generally, the experiments compute aggregates across the entire group, and some distinction between different demographic groups is made only in the Discussion section. The population is grouped only based on likes and followers, not on any other demographic criteria.
It would have been useful to at least have some more data on the demographics found in the posters in the dataset, and the issue of controlling for demographic/cultural criteria mentioned in the Discussion or Future Work sections.

The text processing methods might not be powerful enough to measure the emotion metrics properly, since, as far as I understand, they are based on lexicons and the analysis is done at word unigram level.
Some more details regarding preprocessing methods are necessary: how was the text tokenized, what was done about emojis and punctuation, what other tokens were removed, how big is the vocabulary? These are useful especially given that the linguistic analysis is mostly vocabulary based.



**Questions For The Authors:**

In relation to the ANOVA test: why not actually test for normality instead of assuming it, at least by "manual" inspection of the scores histograms? (which are already illustrated in the paper)

**Reasons To Accept:**

- Interesting goal and some useful results. Novel notion of "biosocial marker" well justified, relatively new metrics of emotion dynamics from text used in relation to mental health diagnosis
- Many different disorders included
- Clear and convincingly written, especially the Introduction
- Sound and thorough statistical analysis

**Reasons To Reject:**

- Stated goal of using emotion dynamics as "biosocial markers" is insufficiently addressed via the methodology, which doesn't focus on the social aspect almost at all
- Text preprocessing details not sufficiently explained
- Potential weakness, but not a serious reason to reject: text processing methods might be too weak to measure emotion dynamics correctly

**Reproducibility:**

3: Could reproduce the results with some difficulty. The settings of parameters are underspecified or subjectively determined; the training/evaluation data are not widely available.

**Reviewer Confidence:**

4: Quite sure. I tried to check the important points carefully. It's unlikely, though conceivable, that I missed something that should affect my ratings.

**Typos Grammar Style And Presentation Improvements:**

- The acronyms in the abstract are only introduced after
- Line 199: "As well..." - incomplete sentence
- Lines 203-206 are unclear to me
- Line 250: "negatively correlated" is meant to be "positively correlated"?
- Footnote on page 6: "small" effect defined as two different intervals
- Line 525: "diagnoses"
- Figure 1 caption: Same caption for subfigures (a) and (b) ?
- Line 647: "aspect" -> "aspects"

---

> ### Author Rebuttal · Authors · 2023-08-28
>
> Dear reviewer,
>
> Thank you very much for your feedback and thorough review.
>
> Re: Stated goal of using emotion dynamics as "biosocial markers" is insufficiently addressed via the methodology, which doesn't focus on the social aspect almost at all
>
> We fully agree with you that more can be done regarding the social aspects of biomarkers. We would like to clarify that the primary goal of this work is to assess the utterance emotion dynamics in language used by various populations associated with mental health conditions. That said, regardless of the extent to which social aspects are experimented with in a work, we think it is always important to refer to such linguistic markers as biosocial. Not treating them so has often, in the past, led to over generalized claims and one-size fit all solutions, at the expense of marginalized communities. Thus, our introduction situates the description of UED markers firmly as bio social. We do not make any claims that this work substantially covers social aspects associated with UED. We will make this clear in the revised version. We also point out below some challenges with addressing social aspects responsibly. Further, we did not want to force-fit more aspects when there wouldn't be adequate space to describe them. We hope for more careful and thoughtful work in this space in the future.
>
> Controlling for social aspects:
>
> In our work, we control for popularity measures on social media as one such aspect. Some other aspects worth exploring include demographics such as age, gender, region, and culture as you are mentioning. We could control for gender, however this has become a sensitive topic as of late. Twitter’s current policy states:
>
> “You should be careful about using Twitter data to derive or infer potentially sensitive characteristics about Twitter users. Never derive or infer, or store derived or inferred, information about a Twitter user’s: Health (including pregnancy), Negative financial status or condition, Political affiliation or beliefs, **Racial or ethnic origin**, or beliefs, Sex life or sexual orientation, Trade union membership, Alleged or actual commission of a crime.”
>
> Re: Text processing approach may be too weak:
>
> While ML methods often outperform the lexicon approach at the instance level, in tasks where information is aggregated from many instances (as is the case in generating emotion arcs) the lexicon approaches perform so well that the improvement from ML methods is very small. By obtaining emotion information from many instances in a bin, past work (Teodorescu and Mohammad, 2022 and in Öhman, 2021) has shown that lexicon approaches obtain highly accurate emotion arcs. For example, Teodorescu and Mohammad (2022) show through experiments on 40+ datasets that predicted emotion arcs with these methods obtain ~98.5% correlation with gold arcs. More importantly, with this approach, we are able to optimize interpretability and ease of computing (no training data or heavy compute power is needed).
>
> Teodorescu, Daniela, and Saif M. Mohammad. "Frustratingly easy sentiment analysis of text streams: Generating high-quality emotion arcs using emotion lexicons." arXiv preprint arXiv:2210.07381 (2022).
>
> Öhman, E., 2021, December. The validity of lexicon-based sentiment analysis in interdisciplinary research. In Proceedings of the Workshop on Natural Language Processing for Digital Humanities (pp. 7-12).
>
> Text processing details:
>
> We have added into the main text details on the text processing in depth. Extra figures and diagrams will be added to the appendix as well. Thank you for pointing it out.
>
> We have also included a formal test for normality in the paper and inspected the data through plots.
>
> Thank you for sharing the reference to "An emotion and cognitive based analysis of mental health disorders from social media data". We have added it in.
>
> We thank you for your thoughtful review and engaging with our work. We see the time and thought you have put into this and very much appreciate it.

---

### Meta-Review · Area_Chair_xGyA · 2023-09-18

**Recommendation:** 5

**Metareview:**

This paper studies the relationship between mental health disorders and social media post emotion dynamics (tweets). The work highlights correlations between mental health diagnoses and linguistic cues about emotion dynamics. The reviewers are in agreement that the topic is interesting and that the work is generally sound. The reviewers' concerns seem to have been addressed during the discussion period.

---

### Decision · Program_Chairs · 2023-10-07

**Decision:**

Accept-Main

**Comment:**

This paper studies the relationship between mental health disorders and social media post emotion dynamics (tweets). The work highlights correlations between mental health diagnoses and linguistic cues about emotion dynamics. The reviewers are in agreement that the topic is interesting and that the work is generally sound. The reviewers' concerns seem to have been addressed during the discussion period.